

# Chatbots integrated in the metaverse: a conceptual model based on interaction ritual chain theory

Yan Yan[1] and Mengjuan Fan[2]

[1] Faculty of Data Science, City University of Macau, Macau, China
[2] Culture, Media, Tourism and Sports Bureau of Jiangmen Municipality, Jiangmen, China

## ABSTRACT

Chatbots are increasingly prevalent as conversational agents in the metaverse, enabled by commercial potential and artificial intelligence (AI) advancements that facilitate natural human-computer interaction (HCI). However, integrating chatbots as virtual community members remains challenging due to deficiencies in conversational and social intelligence. Current metaverse research prioritizes foundational infrastructure while neglecting commonalities between chatbots and community members. This gap is evident in the absence of standardized models for chatbots' social capabilities. To address this, we propose a conceptual model of chatbots' social capabilities based on the interaction ritual chain (IRC) theory, derived from a systematic analysis of 1,273 articles. Our qualitative methodology synthesizes 83 references, all of which dealt with the topic of chatbots and had direct or potential implications for the metaverse. The model addresses core challenges through four elements—virtual coexistence, dynamic constraint, shared intention, and positive incentive—providing solutions for chatbot integration. Finally, we examine social capability interrelationships and suggest future research directions.

# INTRODUCTION

## Chatbots and metaverse

Chatbots, which are computer programs employing artificial intelligence (AI) to interpret customer inquiries and automate conversational responses, have gained renewed prominence following ChatGPT's emergence. This resurgence is driven by two developments. First, large language models (LLMs) have transformed work, leisure, and social interactions (*Zan et al., 2023*), making chatbots indispensable in the metaverse. Users increasingly rely on natural language interfaces for human-computer interaction (HCI), navigating virtual spaces as search engines guide web exploration. Second, technological advances enhance chatbots' human-like qualities. Digital-human systems enable 3D real-time rendering with motion capture, while augmented reality/virtual reality (AR/VR) and haptic technologies provide visual-tactile immersion during interactions. Consequently, chatbots have evolved into integral virtual community components. For this survey, we define chatbots as non-embodied social agents facilitating natural HCI in the metaverse, supporting both general and task-specific community activities.

Corresponding author
Yan Yan,
d22092100108@cityu.edu.mo

The metaverse has evolved significantly in recent years, developing interoperable, immersive virtual communities accessible through user avatars (*Xu et al., 2023*). Current iterations feature diverse chatbot applications. In multiplayer online games, chatbots enhance player engagement by improving non-player characters (NPCs)' narrative contributions to storylines. Platforms like Roblox—a user-generated 3D community with over 80 million daily active users—enable chatbot creation *via* Google or OpenAI application programming interface (API). At its 2023 Developers Conference, Roblox announced conversational AI assistants to streamline experience creation. Chatbots serving as digital sales avatars also enhance personalized shopping on virtual commerce platforms, exemplified by Amazon's 'Rufus' virtual shopping assistant. Additional applications span education (*Kuhail et al., 2023*), healthcare (*Tam et al., 2023*), and beyond.

While chatbots show promise in the metaverse, their development remains nascent. The metaverse accommodate massive concurrent users, chatbots, and entities cohesively. For instance, 3D multiplayer online games frequently host over 100 entities per session. The metaverse, as conceptualized, necessitates seamless social dynamics among multiple agents that mirror the structural characteristics of real-world social systems. Current chatbots predominantly operate in one-on-one interactions, struggling to maintain consistent, continuous dialogues. Effective chatbots require real-time metaverse feedback to assess conversation states, yet their deep involvement in virtual community management remains limited industrially.

## Research question

This study examines the challenges encountered by chatbots operating as members of virtual communities within the metaverse and explores potential solutions. As an emerging domain with diverse demands, the metaverse has garnered growing scholarly interest. While some conceptualize it as a virtual mapping of the real world, the inherent complexity of real-world social systems renders the development of accurate mathematical models or technical indicators for short-term goals a daunting task. Consequently, establishing a standardized framework to guide chatbot design poses significant challenges. These challenges are further exacerbated by the fact that, unlike shared technological infrastructures, chatbots inherently possess social attributes within the metaverse. Understanding the complexities of user-chatbot interactions thus requires an interdisciplinary approach. Moreover, the integration of chatbots into the metaverse remains in its infancy, necessitating further forward-looking technological exploration.

Employing an analytical framework rooted in the interaction ritual chain (IRC) theory, this research evaluates the social capabilities of chatbots. The IRC theory offers a robust framework for analyzing the interaction mechanisms among participants in virtual communities. Its core components—bodily co-presence, barriers to outsiders, mutual focus of attention, and shared mood—interact to form rituals, which in turn generate group solidarity, individual emotional energy, symbols of social relationships, and moral norms.

**Table 1 Comparison of key focuses: related chatbot-specific surveys *vs*. this study.**

| Reference | A key focus of the survey | How our survey differs |
|---|---|---|
| *Huang, Zhu & Gao (2020)* | Addressing three core challenges (semantics, consistency, interactiveness) in open-domain chatbots and summarizing supporting techniques. | Our survey focuses on chatbots in the metaverse, exploring the social capabilities that chatbots should possess to become members of virtual communities, and organizes existing technical achievements from an interdisciplinary perspective. |
| *Wang et al. (2024a)* | Comprehensive survey on LLM-based agents, covering construction, multi-field applications, evaluation, and key challenges. | |
| *Xi et al. (2025)* | Exploring LLM-based agents' philosophical origins, general frameworks, and future directions like artificial general intelligence. | |
| *Ng & Zhang (2025)* | Systematic review of trust in AI chatbots, including conceptualization, influencing factors, and outcomes. | Rather than focusing on how to use chatbots, we pay more attention to how to implement chatbots in the metaverse so that they can establish stable interactive relationships with human users. |
| *Kurniawan et al. (2024)* | Evaluating AI chatbots for chronic illness management, focusing on user acceptance and health outcomes. | |
| *Yusuf, Money & Daylamani-Zad (2025)* | Analyzing pedagogical AI agents in higher education, developing frameworks for educational applications. | |
| *Senadheera et al. (2025)* | Reviewing chatbot adoption in local governments, including purposes, benefits, and implementation frameworks. | |

This study aims to enhance researchers' and designers' understanding of chatbots' social capabilities and identify solutions to existing challenges. The establishment of such standards could facilitate a clearer grasp of objectives, challenges, and enabling technologies, thereby potentially boosting user trust in chatbots.

## Related works

Recent years have witnessed significant efforts to enhance chatbots for user experiences, accompanied by several related surveys. Table 1 summarizes these surveys and contextualizes the value added by our work.

*Huang, Zhu & Gao (2020)* established foundational insights by addressing three core challenges in developing open-domain chatbots—semantics, consistency, and interactivity—and summarizing key techniques (retrieval-based, generation-based, and hybrid methods) that underpin such systems. Building on this, recent surveys have expanded focus to LLM-based agents: *Wang et al. (2024a)* offered a comprehensive overview of LLM-based autonomous agents, covering their construction (encompassing profiling, memory, planning, and action modules), applications across social science, natural science, and engineering, evaluation methods, and key challenges (*e.g.*, role-playing capability and hallucination); similarly, *Xi et al. (2025)* examined these agents from philosophical origins to a general framework (brain, perception, action), their applications in single-agent, multi-agent, and human-agent scenarios, and agent societies, and highlighted synergies between LLM and agent research, evaluation dimensions, and future directions (*e.g.*, paths to artificial general intelligence and Agent as a Service).

Beyond technical advancements, cross-disciplinary research has explored how chatbots foster user relationships across various contexts. *Ng & Zhang (2025)* conducted a systematic review of trust in AI chatbots, identifying user-, machine-, interaction-, social-, and context-related factors influencing trust, with outcomes spanning affective, relational, behavioral, cognitive, and psychological domains, and emphasizing the need for longitudinal studies to understand trust dynamics. In healthcare, *Kurniawan et al. (2024)* reviewed randomized controlled trials on AI-powered chatbots for chronic illness management, noting high user acceptance and partial effectiveness in improving health outcomes. In education, *Yusuf, Money & Daylamani-Zad (2025)* analyzed 92 studies on pedagogical AI conversational agents in higher education, developing a framework that categorizes their pedagogical applications and technological functions, and identifies opportunities to enhance assessment and personalization. In public administration, *Senadheera et al. (2025)* reviewed chatbot adoption in local governments, identifying common purposes (*e.g.*, information provisioning), benefits (*e.g.*, enhanced engagement), risks (*e.g.*, ethical concerns), and influencing factors (*e.g.*, perceived humanness), and proposing a framework to inform implementation. Collectively, these studies demonstrate the evolving role of chatbots as versatile tools, with their impact shaped by technical advancements, trust dynamics, and context-specific needs across disciplines.

Although the technical aspects of implementing the metaverse have been extensively reviewed in the literature, there is a lack of a unified framework to guide the integration of chatbots into the metaverse as virtual community members. *Park & Kim (2022)* outlined the components and approaches necessary for implementing the metaverse, including hardware, software, content, user interaction, implementation, and applications. *Tang et al. (2023b)* focused on delineating the roadmap for the metaverse concerning communication and networking in sixth-generation (6G) wireless systems, integrating technologies such as intelligent sensing, digital twins (DT), space-air-ground-sea integrated networks, multi-access edge computing, and blockchain into 6G. *Xu et al. (2023)* emphasized the convergence of AI and blockchain, which can facilitate service delivery for an edge-enabled metaverse. Additionally, *Wang et al. (2023b)* discussed privacy invasions, security breaches, and proposed research directions for future metaverse systems. These surveys primarily review and discuss research centered on key implementation technologies and applications. Our work, by contrast, aims to direct designers' attention to chatbots, focusing on their integration into the metaverse to advance virtual community development.

Next, the research is structured into five sections: background and methodology, result 1: conversational intelligence, result 2: social intelligence, discussions, and conclusions. The background and methodology section provides a detailed explanation of the systematic analysis inspired by the IRC theory, focusing on the interconnected components between chatbots and virtual community members in the metaverse. The review process employed a retrieval strategy that ensures comprehensive and fair procedures, thereby avoiding any imbalanced assessment that might favor particular interpretations or perspectives. The result sections offer insights into the conversational and social intelligence of chatbots, as well as their challenges and prospective solutions.

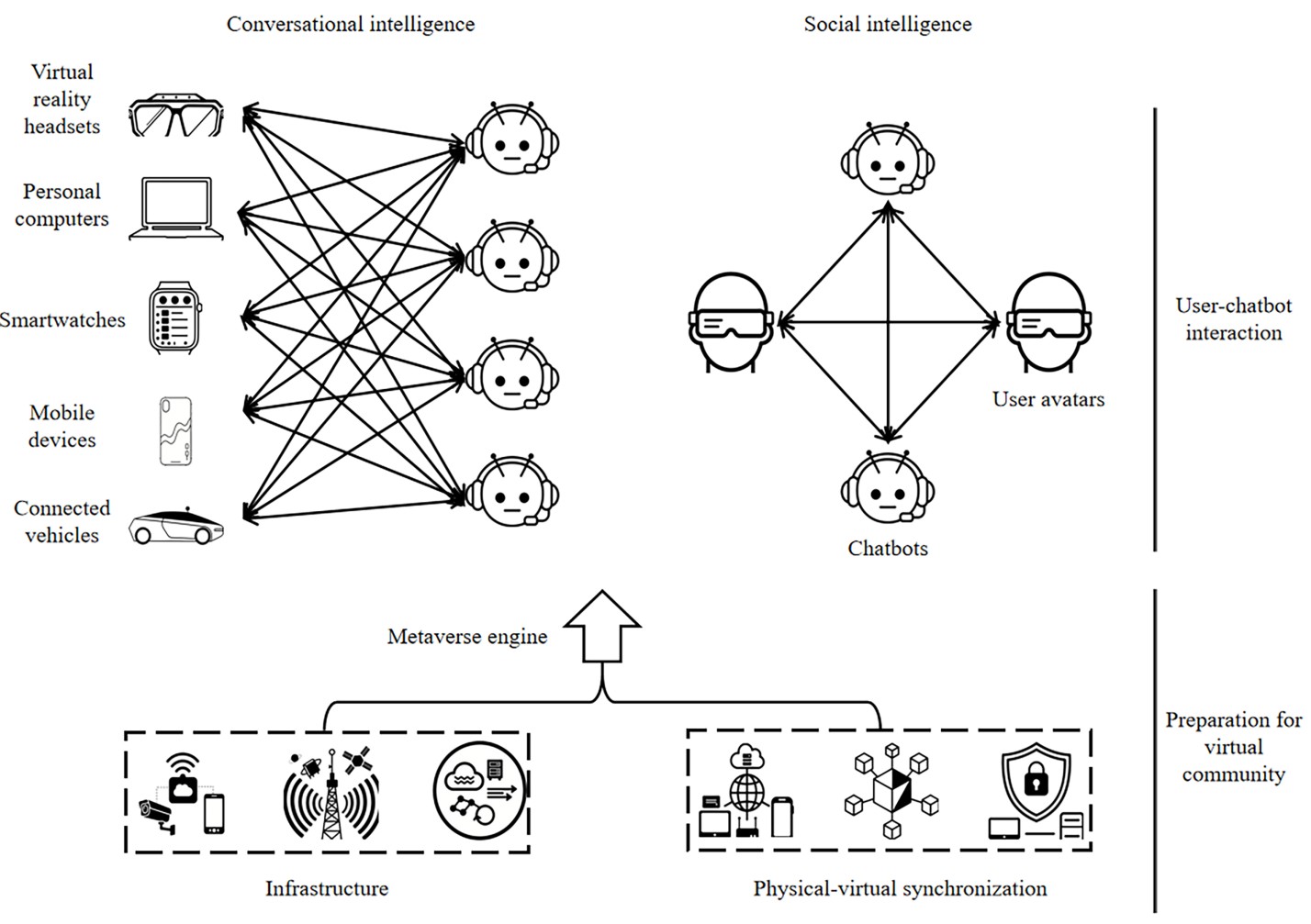

**Figure 1 Chatbot integration in the metaverse.** The foundational infrastructure enabling this integration is represented by icons corresponding to sensing, communication, storage, Internet, blockchain, and cybersecurity technologies. Positioned centrally, chatbots leverage conversational intelligence to interact with users *via* diverse devices, including VR headsets, personal computers, smartwatches, mobile devices, and connected vehicles. Additionally, their interactions with digital avatars exemplify social intelligence, underscoring the role of chatbots in facilitating and reinforcing virtual community construction within the metaverse.

These findings enable designers to make informed decisions by selecting the appropriate subset of social capabilities and inspire further investigations by researchers. The discussions section elaborates on the significance of these findings, emphasizing the importance of social capability interrelations and future directions in chatbot design. Finally, the research concludes with a section on conclusions.

## BACKGROUND AND METHODOLOGY

### Chatbots integrated in the metaverse

The metaverse comprises virtual communities built on advanced Internet technologies that enable shared activities and emotional engagement. As shown in Fig. 1, virtual community construction involves two stages: preparation and user-chatbot interaction. Key infrastructure supports the metaverse, including: parallel sensing, ubiquitous

cognition, scalable storage, and real-time analysis. These elements sustain its dynamic interactivity. Physical-virtual synchronization relies on foundational technologies: AR/VR (immersive experiences), integrated sensing-communication (seamless interaction), DT (virtual replication), AI (intelligent automation), and online social networks (social connectivity). Collectively, these components empower member-driven protocols for cohesive, responsive virtual communities.

Furthermore, these communities emerge through effective user-chatbot interactions, generating group cohesion, individual emotional engagement, shared social symbols, and collective ethical standards. Despite differing objectives and characteristics between chatbots and community members, fundamental commonalities establish robust bonds and mutual benefits. Successfully leveraging chatbots for virtual community construction hinges critically on user willingness to form stable IRCs with chatbots.

Chatbot intelligence critically influences both experience design and user engagement. Consequently, an interdisciplinary perspective—widely applied in virtual community research—can elucidate interaction-relevant intelligence types and augment chatbot capabilities. Designers must consider how chatbot behavior affects user interactions (including metaverse integrations), user experience, and participation levels. While HCI offers valuable development insights (*e.g.*, manipulation techniques, interface design, and behavioral patterns), chatbot design complexities inherently transcend traditional HCI boundaries.

## Chatbots and virtual community members: interconnected elements
### Conversation
Conversation serves as a foundational component for chatbots and virtual community members. Guided by five dialogic principles—mutuality, propinquity, empathy, risk, and commitment—these interactions establish frameworks for high-level conversation (*Kent & Taylor, 2002*; *Jiang et al., 2022*; *Kent & Taylor, 2021*). High-level conversations feature: risk-taking, altruistic information exchange, turn-taking respect, inclusive ethical deliberation, rhetorical sensitivity to others' needs, and participant-adapted communication. This distinguishes them from basic bidirectional exchanges. Since the Turing test (*Turing, 2009*), AI conversational systems have evolved substantially. *Nißen et al. (2022)* classifies chatbot engagement into three categories: *Ad-hoc* Supporters, Temporary Advisors, and Persistent Companions. Designed for longitudinal relationships, persistent companions demonstrate: autonomous adaptation, advanced personalization, anthropomorphic design, socio-emotional behaviors, and socially oriented communication. Their experiential learning and model optimization intrinsically satisfy high-level conversational criteria, especially inclusiveness.

The primary objective of conversational intelligence is maintaining persistent user engagement. This necessitates chatbots with social capabilities that enhance human-agent co-presence while implementing dynamic constraints to mitigate external interference during interactions. User-centric design prioritizes three core functionalities in chatbots: initiative support, responsibility management, and communication assistance, enabling

seamless the metaverse access. Beyond content delivery, high-level conversational chatbots must exhibit behavioral inclusiveness through multimodal information fusion, knowledge orchestration, and cross-domain adaptation.

### Sociability

Beyond providing reliable and personalized conversational intelligence, user-centric chatbot services require sophisticated social intelligence. The social intelligence are fundamentally collaborative in virtual communities. Within the metaverse, members function as autonomous agents that interact to achieve shared or individual goals. These agents may include software programs, intelligent devices, or human users. Metaverses constitute complex cyber-physical-social systems that integrate virtual and real-world socio-economic ecosystems (*Zhang et al., 2018*; *Wang et al., 2022*), with chatbots functioning as critical sociotechnical interfaces. Their deployment in university platforms exemplifies how personalized, immediate support capabilities enhance user experience (*Xie et al., 2023*). Chatbots' socially assistive roles—such as supporting isolated populations, neurodiverse individuals, and psychologically vulnerable groups (*De Gennaro, Krumhuber & Lucas, 2020*; *Valtolina & Marchionna, 2021*; *Xygkou et al., 2024*; *Ali et al., 2020*; *Zhou et al., 2020*)—demonstrate significant societal value. Thus, integrating social intelligence into chatbots requires IRC-aligned designs to foster shared emotional states and identity formation. This integration involves three key actions: evaluating chatbots as ritual participants (*Sun, Zhan & Such, 2024*); mitigating the limitations of LLMs; and implementing cooperative multi-agent reinforcement learning (MARL) systems, where chatbots evolve through environmental interaction to sustain mutual benefits.

Applying the IRC theory to virtual communities provides a critical framework for analyzing evolving social intelligence. Human-agent interaction has fundamentally transformed community ownership and interaction paradigms (*Qiqi, 2023*; *Simons, 2019*), necessitating evaluations that go beyond content analysis to include longitudinal engagement, emotional resonance, and sensory effects—key dimensions for enhancing sociability in the metaverse. To understand the social intelligence involved in facilitating user-chatbot interactions and advancing chatbot development, we examine the IRC theory. This framework helps elucidate the key components that foster the emergence of shared emotions and behaviors during virtual social interactions. Table 2 presents a comparative analysis of the elements in our chatbot model and the IRC theory.

## Survey methodology

The number of review articles published in computer science-focused journals has remained relatively stable, a trend that reflects the enduring impact of such research. Our systematic review aims to synthesize and analyze evidence related to research questions about the social capabilities of chatbots. This study follows the Preferred Reporting Items for Systematic Reviews and Meta-Analyses (PRISMA) guidelines to enhance the transparency and rigor of the literature review, as shown in Fig. 2.

**Table 2 Mapping from four elements of IRC to that of chatbots.** The first column provides an overview of chatbots' social capabilities, which include conversational intelligence and social intelligence. The subsequent columns delineate the IRC and chatbot models, respectively. Each row contrasts methods from both domains that aim to achieve related short-term goals. For instance, bodily co-presence in the IRC theory emphasizes monitoring participants for interaction, while the chatbot model focuses on gathering user information for effective state synchronization.

| Capability | IRC | Chatbots | Objective |
|---|---|---|---|
| Conversational intelligence | Bodily co-presence | Virtual coexistence | Chatbots and users can be in the same situation to influence each other. |
| | Barriers to outsiders | Dynamic constraint | Chatbots can determine the division of member groups to avoid outsiders' disturbances. |
| Social intelligence | Mutual focus of attention | Shared intention | Chatbots can collaborate with users to ensure that the group's common focus remains condensed. |
| | Shared mood | Positive incentive | Chatbots can form a system with other intelligent agents, often with long-lasting effects. |

### Eligibility criteria and search strategy

To identify the most relevant databases, preliminary searches were conducted across multiple platforms, with relevance evaluated based on the number of retrieved articles and their alignment with the research questions.

An extensive review of existing publications on "chatbots integrated in the metaverse" revealed a scarcity of literature directly addressing the research question, largely attributable to the topic's novelty. Consequently, this study initially identified empirical studies and prior research potentially relevant to answering the research question. Several databases and search engines were employed, including ScienceDirect, Taylor & Francis Online, Wiley Online Library, IEEE Xplore, Springer, Google Scholar, and arXiv. Google Scholar and arXiv served as supplementary resources due to their coverage of interdisciplinary research and emerging technologies in recent years.

Filters applied during literature searching included: scientific articles, English language, and no initial date restriction. However, this review focused on articles published between 2018 and 2025 (directly or indirectly related to chatbot technologies and the metaverse), as elaborated in "Reports excluded, reason 1".

Search keywords included "chatbot and metaverse," "chatbot in metaverse," "chatbot role in metaverse," "chatbot applications in metaverse," and "chatbot challenges for metaverse." Given the extensive body of research on chatbots in computer science, additional keywords such as "conversational agent," "conversational model," "dialogue system," and "social agent" were incorporated to capture relevant literature. Notably, metaverse development is closely intertwined with advancements in AI; accordingly, research hotspots including "large language model," "human-agent interaction," and "multi-agent system" were included. Both chatbots and the metaverse have emerged as interdisciplinary research hotspots; however, this review specifically focuses on AI and HCI within computer science. Publications outside this scope were excluded during initial screening. The initial search yielded 1,273 articles.

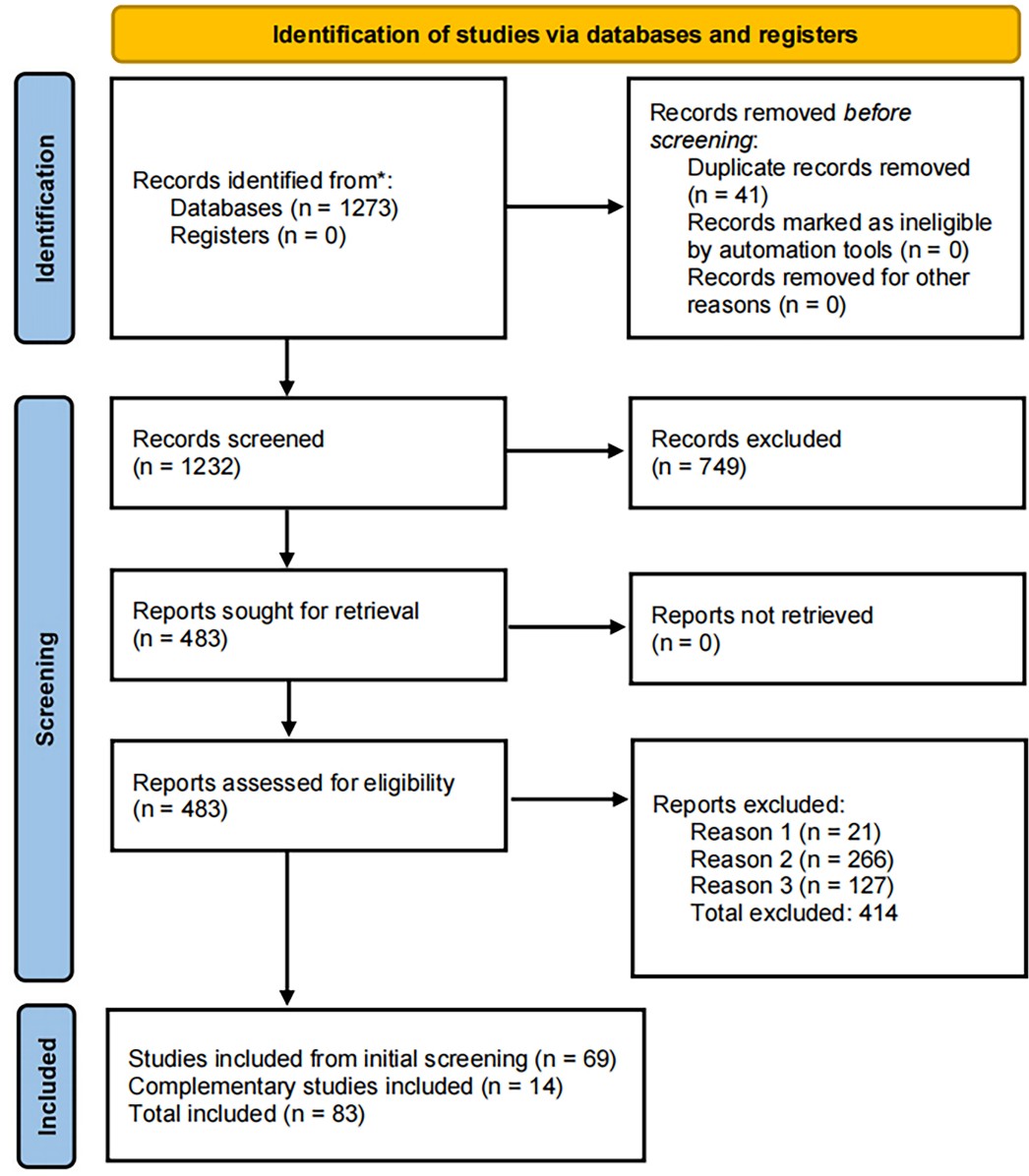

**Figure 2 PRISMA flow chart process of article selection.**

## Screening

Article selection was guided by keywords and abstracts, with each manuscript assessed for relevance to the research question and academic impact. The screening process adhered to PRISMA guidelines.

Duplicate removal: A total of 41 duplicate articles were identified and removed using reference management software (*e.g.*, Zotero), resulting in 1,232 unique articles.

Title and abstract screening: A total of 749 articles were excluded after screening titles, abstracts, and keywords for misalignment with core topics (metaverse, chatbot, natural

language processing (NLP), multi-agent systems, virtual world), leaving 483 articles for full-text assessment.

- Reports excluded, reason 1: ($n = 21$) articles published before 2018. Although no initial date filter was applied ("anytime"), subsequent analysis determined that studies included in the systematic literature review should be restricted to the preceding 8 years (2018–2025). This timeframe aligns with the emergence of modern chatbot frameworks (*e.g.*, transformer-based seq-to-seq models post-2018) and the growing prominence of the metaverse, as noted by *Dwivedi et al. (2022)*, who highlight increased scholarly attention to the metaverse after 2020—including Facebook's rebranding to Meta in 2021.
- Reports excluded, reason 2: ($n = 266$) irrelevance to the research topic. These included articles mentioning "metaverse" but outside computer science (*e.g.*, pedagogy) or addressing the topic without providing substantive insights (*e.g.*, metaverse infrastructure).
- Reports excluded, reason 3: ($n = 127$) misalignment with the research objective. These studies focused primarily on hardware-software integration in robotic systems rather than chatbot programming for virtual community development.

Supplemental searching: Conducted in response to peer reviewer feedback indicating omissions of literature on technical security aspects of deep machine learning and federated learning—both prevalent in chatbot training and deployment, particularly in metaverse applications. No additional duplicates were identified in supplemental searches.

### Selection

- Reports included, reason 1: ($n = 16$) articles aligned with the topic, explicitly addressing "chatbot," "metaverse," or "HCI," or focusing on relevant technologies (*e.g.*, AR/VR, NLP, HCI) despite lacking explicit mention of core terms.
- Reports included, rReason 2: ($n = 53$) articles on chatbots not centered on the metaverse but applicable to virtual community research.
- Reports included, reason 3: ($n = 14$) articles from supplemental searches, focusing on deep machine learning, federated learning, and technical security.

Final inclusion: For synthesis, 83 articles were included and analyzed through the lens of the chatbot conceptual model, as shown in Fig. 3.

## RESULT 1: CONVERSATIONAL INTELLIGENCE

### Virtual coexistence

Virtual coexistence aims to intelligently analyze social dynamics and generate contextually appropriate responses—mirroring IRC's bodily co-presence principle—to enhance interaction success through comprehensive user profiling. Face-to-face engagement and communal spaces are essential for boosting participative comprehension and interactivity. However, technological constraints severely limit implementation: unlike conventional voice/text dialogues, metaverse chatbots must engage diverse entities in open-domain

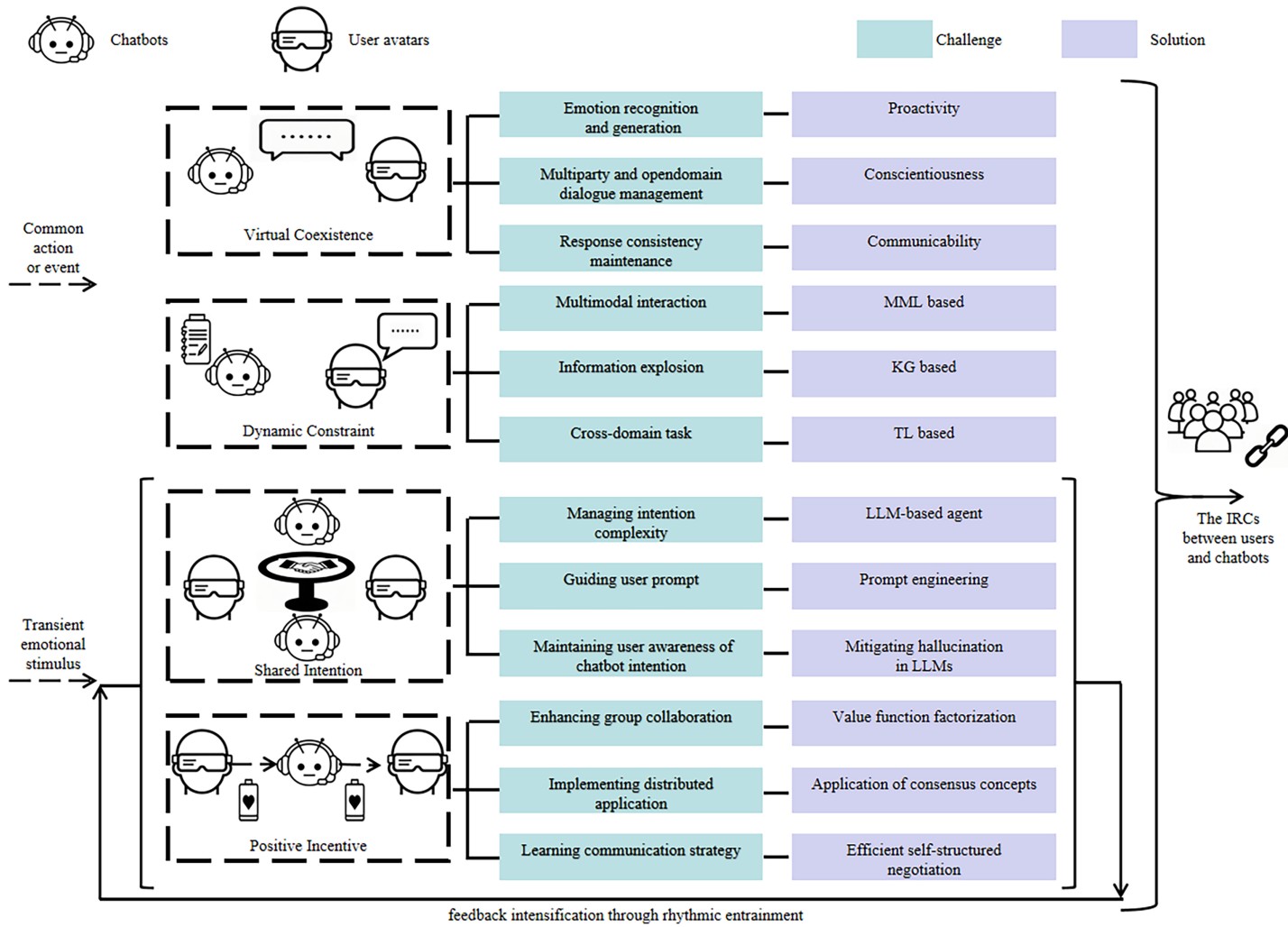

**Figure 3  The model of interaction ritual between users and chatbots.** Common action or event, as well as transient emotional stimulus, drive the initiation of interactive rituals. The four social abilities of chatbots facilitate the development of these interactive rituals in the metaverse. The interaction between users and chatbots in virtual coexistence, combined with dynamic constraints, provides optimal conditions for users to share intentions and transmit positive incentives. This, in turn, generates moments of intersubjectivity that produce the outcomes of IRCs between users and chatbots.

contexts and decode complex social presence indicators (*e.g.*, similarity, familiarity, attraction) from multimodal inputs. These requirements exceed current conversational intelligence capabilities, presenting fundamental implementation barriers.

### Challenges of virtual coexistence

[C1] Emotion recognition and generation: Ineffective emotion recognition and generation jeopardize virtual coexistence success. Current techniques face difficulties in contextual modeling, capturing conversational dynamics, and representing speaker states. Chatbots with deficient emotion recognition produce generic responses, analogous to humans offering non-committal replies when uncertain. Furthermore, traditional neural language models struggle to generate logically coherent and emotionally resonant personalized

responses. Simply mirroring user sentiments without providing actionable guidance causes message neglect and impairs emotion generation capabilities.

[C2] Multiparty and open-domain dialogue management: Chatbots must sustain efficient goal-oriented interactions in multiparty open-domain dialogues. A restricted response repertoire may indicate inattentiveness to user inputs. Distinguishing nuanced variations across similar dialogue compositions remains an unresolved challenge. Ensuring structured conversational flow while masking multiparty constraints requires extensive training on addressee-annotated datasets. Users frequently evaluate chatbot membership suitability based on response diversity and generalization capability. Although multi-turn interactions are essential for complex goal attainment, they increase context coherence fragility.

[C3] Response consistency maintenance: Inconsistent chatbot responses can cause user misinterpretations and impair communicability. Unlike traditional software interfaces, chatbots articulate functionality and interaction principles through multi-turn dialogues. Their semantic analysis engines—which convert user utterances into executable commands—are critical components. However, multilingual models show reduced reliability for semantic analysis. While natural language effectively defines tasks and describes diverse entities, current chatbots face challenges integrating vision-language models, particularly when interpreting complex visual scenarios. Ensuring fairness and inclusion is essential for communicability; neglecting these principles impedes common ground establishment, potentially destabilizing virtual coexistence.

### Solutions of virtual coexistence

[S1] Proactivity: Proactivity enables chatbots to actively engage users and enhance conversational flow through initiating dialogues, proposing topics, supplementing information, and asking follow-up questions. For emotion modeling, *Lei et al. (2023)* introduced emotional cognitive loss and hybrid continuous attributive networks to reduce speaker modeling overfitting, improving cross-scenario performance. *Ranasinghe et al. (2022)* developed an NLP-acoustic framework for detecting emotional transitions in telehealth. *Sun, Li & Tao (2022)* proposed a syntax-constrained bidirectional framework integrating emotion/topic keywords to generate contextually coherent responses.

Chatbots also must balance comfort provision with informational content, particularly for distressed users. *Deng et al.'s (2023)* knowledge-enhanced framework enables proactive mental health dialogues. *Tu et al.'s (2022)* strategy-aware model infers emotional states to deploy multi-tactic support. Addressing prior limitations, *Cheng et al.'s (2022)* multi-turn system detects nuanced emotional cues and employs A*-inspired lookahead heuristics for optimal strategy selection.

[S2] Conscientiousness: Conscientiousness enables chatbots to achieve effective virtual coexistence in the metaverse. This attribute allows comprehensive contextual understanding of utterances and autonomous recovery from operational failures. Unlike dyadic dialogues, multiparty conversations require responses predicated on recipient role prediction. *Liu et al.'s (2023e)* pseudo-variational hierarchical model identifies subtle semantic variations without prior knowledge, synthesizing variables optimized *via*

distribution consistency objectives. *Lv et al.*'s *(2023b)* hierarchical duality learning emulates human cognition in inferring utterances from logical structures. To address limited multiparty pretraining data, *Li & Zhao (2023)* proposed expectation maximization for iterative addressee annotation and response optimization.

In open-domain dialogues, chatbots achieve generalization through many-to-many mappings where single contexts elicit diverse responses. *Zhao et al. (2023)* introduced robust metrics for one-to-many evaluation. For multiparty contextual coherence, *Lv et al.*'s *(2023a)* dialogue path-sampling projects conversations onto extended Brownian Bridges, sampling latent variables to construct coherent paths that enrich training data.

[S3] Communicability: Communicability constitutes the third critical dimension for chatbots, particularly vital in the metaverse's multilingual environments. Chatbots must perform robust multilingual semantic analysis to accommodate users who frequently switch languages. *Held et al.*'s *(2023)* doubly aligned multilingual parser significantly advances zero-shot capabilities while reducing model parameters, enhancing cross-linguistic communication.

*Liu et al.*'s *(2023f)* InternGPT equips chatbots with planning/reasoning capacities through interactive visualization. This framework enables comprehension of non-verbal commands (*e.g.*, gestures toward on-screen content) during vision-centric tasks, bridging visual-linguistic gaps to boost communication efficiency.

*Sicilia & Alikhani (2023)* formalized fairness in text generation *via* computational learning theories. Their work clarifies anthropomorphic learning-fairness relationships, demonstrating that algorithms can produce equitable text from biased data under fairness-preserving conditions.

In virtual communities, active chatbots may number in the hundreds, while the devices they serve can scale to millions. Each chatbot deployment requires distinct adaptations for both devices and users. Federated learning—a distributed paradigm that trains global models using client data without sharing raw data—faces two critical limitations. First, research reveals significant privacy risks, as malicious servers can reverse-engineer model gradients to compromise data (*Mothukuri et al., 2021*). Second, global models often underperform for individual clients when local data exhibit heterogeneity in quantity, quality, or distribution. To address these challenges, *Zhang et al. (2024)* proposed a privacy-preserving, personalized framework that partitions gradient computation between servers and local clients, thereby optimizing models for specific contexts. Complementarily, *Xu et al. (2022)* introduced a method to mitigate accuracy degradation from irregular user participation, ensuring training outcomes primarily reflect high-quality data contributions.

## Dynamic constraint

Dynamic constraint employs computational methods that leverage virtual coexistence data to formulate conversation strategies aligned with virtual community structures, reflecting the IRC theory's Barriers to Outsiders principle. These strategies explicitly differentiate participants from non-participants by utilizing communal space information to reinforce interaction rituals, where accurate identification of active participants is

essential. This capability enables chatbots to distinguish and segregate disengaged members during social interactions. Dynamic constraint perceives multidimensional social boundaries—including temporal and spatial dimensions—to ensure information relevance for community members. Consequently, chatbots initiate context-sensitive interaction patterns that facilitate seamless integration into target virtual communities. This section delves into potential solutions across various research domains, including multimodal machine learning (MML), knowledge graph (KG), and transfer learning (TL).

### Challenges of dynamic constraint

[C1] Multimodal interaction: The influx of multimodal messages poses a significant challenge for chatbots in the metaverse, particularly in restricting unauthorized participants in real-time interactions. User–chatbot interactions involve a complex interplay of various modalities such as speech, non-verbal sounds, visuals, and text. Chatbots face two main challenges in processing this diverse information. The first is synchronizing the different modalities within a message, known as the fusion problem. The second is identifying the implicit content often conveyed through multimodal messages, known as the recognition problem. For instance, face-to-face interactions often involve asynchronous and emotionally charged multimodal messages.

[C2] Information explosion: Dynamic constraint necessitates managing diverse information, which may compromise technical performance. Effective chatbots require strategies to distill relevant content from extensive data—a core aspect of dynamic constraint. Knowledge management system development remains complex due to intricate representation and reasoning requirements. As virtual community knowledge dynamically evolves with metaverse applications, chatbots must rapidly acquire and generalize new knowledge. Addressing information overload through concise abstractions and scalable frameworks presents significant technical challenges.

[C3] Cross-domain task: Predicting user behavior for chatbots implementing dynamic constraints in cross-domain tasks presents significant complexity. Chatbots must first navigate socio-contextual factors and task variations across domains before applying constraints. Traditional training methods assume consistent input requirements and social contexts within single domains—an assumption often invalid when tasks exhibit ambiguity or indeterminacy. Consequently, cross-domain training data accessibility becomes critical. Enhancing chatbot robustness for cross-domain tasks is fundamental to successful dynamic constraint implementation.

### Solutions of dynamic constraint

[S1] MML-based solution: These approaches enable chatbots to comprehend and generate video-grounded dialogues. *Wang et al. (2023c)* established benchmarks for video-based dialogue understanding and generation, incorporating multimodal characteristics like scene/topic transitions. *Zheng et al. (2023a)* proposed a contrastive visual-textual transformation method for sign language recognition, leveraging variational autoencoders for contextual pretraining and contrastive alignment for cross-modal consistency.

Simultaneously, *Zhang et al.*'s *(2023a)* quality-aware multimodal fusion framework optimizes information utilization while mitigating low-quality data impacts, and *Lv et al.*'s *(2021)* progressive modality reinforcement addresses misaligned streams through iterative modality-hub exchanges.

For multimodal emotion recognition in multiparty dialogues, visual cues are critical: *Zheng et al.*'s *(2023b)* two-stage framework extracts utterance-aligned facial expressions *via* face recognition/clustering/matching and enhances recognition *via* frame-level emotion distributions; *Liu et al.*'s *(2023a)* brain-machine coupled approach trains on visual-EEG data for cross-domain facial emotion analysis; while *Rasendrasoa et al.*'s *(2022)* knowledge-aware network employs multi-head attention on dialogue history and participant knowledge for joint emotion/dialogue act recognition.

[S2] KG-based solution: KGs serve as collective repositories within organizations, enabling knowledge representation, accumulation, curation, and propagation. Chatbots leverage KGs to refine dialogue generation: *Tang et al.*'s *(2023a)* framework integrates language models with graph operations through multi-stage feature fusion to enhance response informativeness. Furthermore, KG embeddings effectively capture uncertainty and domain-specific knowledge. *Yao et al.*'s *(2023c)* AnKGE framework augments KG embeddings with self-supervised analogical inference, generating analogical object embeddings from pre-trained elements. While neural link predictors identify missing triples in incomplete graphs, they face computational intensity and negation-query limitations. *Wang, Chen & Grau (2023)* addresses these challenges, and *Cui et al.*'s *(2023)* lifelong model transfers knowledge through masked autoencoders and embedding regularization without re-embedding. Alternatively, *Liu et al.*'s *(2023c)* IterDE employs knowledge distillation to compress KG models, improving storage/inference efficiency while outperforming baselines in link prediction.

[S3] TL-based solution: Chatbots can acquire new tasks by drawing on prior experience. Invariance learning techniques help distinguish spurious features from invariant ones within inputs. *Tan et al. (2023)* have proposed an environment-agnostic invariance learning method requiring neither environmental indices nor auxiliary data. Domain generation offers another effective approach, though model efficacy depends on distributional alignment between generated and target domains. *Dai et al. (2023)* introduced moderate distributional exploration for domain generalization, identifying common factors across training domains *via* uncertainty subsets with shared semantics.

Zero-shot domain adaptation challenges chatbot deployment in unseen domains without supervised data. Parameter-efficient TL addresses this limitation. *Aksu, Kan & Chen (2023)* developed the Prompter method, which uses target domain slot descriptions to generate dynamic prefixes interacting with self-attention keys/values per layer, incorporating conversational semantics and co-occurrence frequency. Previous work prioritized data/model-level augmentation for generalization but inadequately isolated sample semantics, constraining zero-shot performance. *Wang et al. (2023a)* proposed a divide-conquer-combine strategy that disentangles observed data semantics *via* mixture-of-experts, enhancing robustness and performance.

Chatbot training employs diverse methods to process multimodal data (images, text, numerical inputs). Formulating cross-modal constraints remains challenging, as user behaviors—varying across temporal sequences and conversational contexts—are critically important for open-domain dialogue generation. Within federated learning paradigms, ethical concerns arise regarding the use of data. Privacy risks persist despite raw data localization: uploaded gradients/parameters may reveal local information, enabling models to infer client data distributions. *Nasr, Shokri & Houmansadr (2019)* demonstrated this *via* membership inference attacks exploiting stochastic gradient descent vulnerabilities. Fairness is equally critical, requiring mitigation of biases against disadvantaged groups (*e.g.*, ethnic minorities, women, the elderly). Such biases originate from both clients (*Jang, Zheng & Wang, 2021*) and servers (*Nishio & Yonetani, 2019*). Violations of privacy or fairness are ethically unacceptable. *Chen et al. (2023a)* surveyed interactions between these principles, revealing their interdependence. Systematic co-optimization prevents sacrificing one for the other.

## RESULT 2: SOCIAL INTELLIGENCE

### Shared intention

Shared intention constitutes the foundational pillar of social intelligence, aligning with the principle of mutual focus of attention in the IRC theory. Chatbots' social roles extend beyond dyadic user interactions, particularly in open-world environments. They are engineered to maintain granular chronicles of social exchanges through natural language, iteratively synthesize these experiences into higher-level cognitive abstractions, and dynamically retrieve such knowledge to optimize behavioral strategies. LLMs prove indispensable in augmenting chatbots' capabilities, transcending the limitations of conventional question-answering systems or recommendation algorithms. However, the inherently complex and expansive nature of shared intention presents fundamental challenges: LLMs' architectural constraints impede chatbots' capacity to manage the continuous evolution of multi-agent intentions, resolve emergent conflicts, track longitudinal event trajectories, and navigate intricate social dynamics arising between users and autonomous agents within collaborative ecosystems.

### *Challenges of shared intention*

[C1] Managing intention complexity: Designing advanced chatbots capable of multi-entity collaboration for complex intention planning, memory enhancement, and action execution remains an open challenge. These chatbots require adaptability and optimization, with core functionalities including: task decomposition, multi-timescale memory management, tool invocation, and multi-agent communication. Although LLMs demonstrate strong general capabilities, many chatbots rely exclusively on short-text inputs and LLMs' native planning/acting functions. This limitation causes stochastic outputs and interaction inconsistencies—a critical challenge requiring further investigation.

[C2] Guiding user prompt: Prompt design critically shapes model outputs (*Lu et al., 2022*). Analogously, LLMs function as chatbot brains where prompts serve as

commands—minor variations yield substantially divergent results. Prompts encompass syntactic elements (*e.g.*, length, blanks, example ordering) and semantic components (*e.g.*, phrasing, example selection, instructions). Prompt quality directly determines LLM task performance, with optimal prompts typically requiring human crafting. When novice users unaware of chatbot limitations issue capability-testing prompts, unmet expectations cause frustration. Consequently, prompt fragility significantly compromises chatbot intention realization.

[C3] Maintaining user awareness of chatbot intention: Chatbots must provide accurate responses for complex intentions. Hallucination—where generated content is illogical or unfaithful to source material—remains prevalent (*Filippova, 2020*). LLMs frequently produce hallucinations containing misleading information, often undetectable due to textual fluency. *Lee et al. (2022)* evaluated these using entity-based metrics and textual entailment. Given the emphasis on user-centric responses and alignment with user directives in LLMs, hallucinations in natural language generation tasks can be divided into factuality hallucination and factual fabrication (*Huang et al., 2025*).

### Solutions of shared intention

[S1] LLM-based agent: Recent advances in LLM-based agents enable users to define virtual agent attributes, conduct dynamic dialogues, and receive automated feedback. *Hasan et al. (2023)* introduced SAPIEN—an advanced platform supporting multilingual communication (13 languages), emotional expression through facial/vocal cues, and non-verbal interaction. Significant progress has also been made in automating complex problem-solving. *Hong et al. (2024)* proposed MetaGPT, a meta-programming framework that integrates human workflows into multi-agent systems. Using an assembly-line approach, MetaGPT delegates roles and decomposes complex tasks into manageable subtasks to enhance agent collaboration.

An LLM-based agent is characterized by the integration of LLMs as the cognitive core, supplemented by several essential components:

- Planning: Language models typically solve general problems across diverse tasks. However, their token-level left-to-right decision-making limits performance on tasks requiring exploration, Strategic lookahead, or critical initial decisions. *Wei et al. (2022)* demonstrated that Chain-of-Thought prompting—providing exemplar demonstrations—enhances LLM reasoning. *Yao et al. (2023a)* extended this approach with the Tree of Thoughts framework, which examines coherent text units ("thoughts") as intermediary problem-solving steps. This enables models to evaluate multiple reasoning paths and strategically select next actions. *Wang et al. (2024c)* proposed Solo Performance Prompting, where a single LLM emulates distinct personas through multi-turn self-collaboration. This method improves complex problem-solving by dynamically adapting to task inputs.
- Memory: The rise of high-performance conversational agents like ChatGPT stems from advances in LLMs. Such agents must retain key dialogue information to generate contextually relevant responses. However, limited user control over memory

management often harms conversational coherence. To tackle this, *Huang et al. (2023)* proposed Memory Sandbox—an interactive system for transparent user oversight of conversational memory. This framework modularizes memories as data objects, enabling viewing, editing, summarization, and sharing. Meanwhile, *Chen et al. (2023b)* utilized LLMs' narrative generation to synthesize artificial memories, converting stories into actionable sequences for guiding agents in hypothetical scenarios. Traditional reinforcement learning methods struggle with LLM-based agents due to costly fine-tuning and sample inefficiency. *Shinn et al. (2023)* introduced Reflexion, a framework that improves language agents *via* linguistic feedback rather than weight updates. Reflexion agents analyze task feedback (*e.g.*, success/failure signals), produce self-critiques, and store them in episodic memory. This iterative refinement achieves 91% pass@1 accuracy on HumanEval, surpassing GPT-4's 80% (previous state-of-the-art).

- Action: While LLMs possess strong generative capabilities, they often struggle with tasks requiring complex planning and external tool usage. To address this, *Liu et al. (2023b)* leveraged a language-only GPT-4 to generate multimodal instruction-following datasets combining text and images. By refining these instructions, they developed a large vision-language assistant—an end-to-end multimodal model integrating a vision encoder with LLMs for general-purpose visual and linguistic understanding. *Yao et al. (2023b)* investigated LLM-based agents' reasoning and action capabilities. Their work focused on using LLMs to generate structured reasoning traces alongside task-specific actions. These traces help the model infer, monitor, and adjust action plans, while actions facilitate interaction with external tools to gather additional information. Further enhancing task planning and tool utilization, *Kong et al. (2024)* proposed a framework with three key components: API retriever, demo selector, and LLM finetuner.

[S2] Prompt engineering: To address prompt brittleness, *Zhou et al. (2023)* proposed an automated prompt engineering system that generates and optimizes instructions. By treating instructions as trainable "programs," their method iteratively refines candidate prompts using LLM-generated suggestions. Experiments demonstrated that this approach enhances few-shot learning performance by improving standard in-context learning prompts. It also effectively identifies high-quality zero-shot chain-of-thought prompts, leading to more accurate and informative model outputs. Complementing this work, *Raman et al. (2023)* introduced model-tuning *via* prompts, a lightweight adaptation method for downstream tasks. The method appends tunable prompt templates to inputs and generates predictions *via* text infilling or completion. When integrated with adversarial training, model-tuning *via* prompts not only preserves performance on clean data but also significantly improves robustness against adversarial attacks.

[S3] Mitigating hallucination in LLMs: A strategy to diminish hallucinations involves Retrieval Augmentation, a method that anchors the model's input to external knowledge sources. *Ge et al. (2023)* augmented the generalization capabilities of language models with mixture-of-memory augmentation, a process that retrieves augmentation documents from multiple information sources, referred to as 'external memories', and permits the addition

of new memory during inference. *Liu et al. (2023d)* developed a hierarchical transformer retriever, specialized in dialogue domain data, for personalized retrieval. They proposed a context-aware prefix encoder to integrate retrieved information effectively. Furthermore, *Zhang et al. (2023b)* introduced a knowledge merging framework oriented towards compatibility, aiming to enhance open-domain question answering systems. This framework posits that answers corroborated by multiple sources are more reliable.

Metaverse applications require diverse model architectures to support heterogeneous training tasks. Consequently, chatbots must integrate multiple features—each associated with unique pre-training tasks and activation conditions. This architectural complexity exacerbates intellectual property risks, particularly model-stealing attacks. For instance, *Carlini et al. (2024)* successfully extracted all projection matrices from OpenAI's Ada and Babbage models for less than $20, proving the feasibility of creating functional replicas. *Hui et al. (2024)* further demonstrated prompt theft through adversarial chatbot outputs. To counter these threats, *Zhou et al. (2024)* proposed an inversion-based defense using neural decoders to analyze suspicious model outputs for legitimacy verification. In parallel, *Li et al. (2023b)* developed generative API watermarking techniques to deter remote imitation attacks.

## Positive incentive

Positive incentives align with the principle of shared mood in the IRC theory, as both generate 'emotional energy'—a pleasurable sense of group affiliation that motivates repeated ritual participation (*Collins, 2005*). In user-chatbot interactions, these incentives function as 'batteries' that store and release interactive energy. They enable experience recall and initiate subsequent interaction rituals, forming interconnected chains. This mechanism addresses inherent complexity from dynamic environments and combinatorial challenges in chatbot operations. The approach prioritizes training models for proficient group collaboration through methodologies like value function factorization, consensus-building techniques, and structured negotiation protocols.

### Challenges of positive incentive

[C1] Enhancing group collaboration: A key challenge in multi-agent systems lies in learning value functions that effectively promote collaborative behavior. Traditional approaches involve either optimizing a joint reward function or reformulating the problem as a single-agent system. However, traditional reinforcement learning methods often fail to reach global optima in large state-action spaces, primarily due to two factors: (1) agents' inability to accurately assess their individual contributions to collective rewards, leading to diminished motivation, and (2) suboptimal policies from certain agents that degrade overall performance and hinder optimal policy discovery. Value function factorization has emerged as a promising solution, enabling precise attribution of each agent's contribution to the global reward. This approach maintains the theoretical advantages of centralized training while facilitating decentralized execution, effectively addressing the credit assignment problem in cooperative multi-agent environments.

[C2] Implementing distributed application: Chatbots must achieve an optimal solution by reaching consensus with their peers. With an increasing number of chatbots, the information they manage could surpass their processing capacity. Furthermore, in the metaverse, centralizing local data onto a central server presents challenges due to energy constraints, privacy issues, and hardware failures. The issue becomes how to facilitate communication among local agents and adjust the network topology to enable them to collaborate effectively and learn the optimal policy. The consensus approach entails structuring agents within a sparse network, allowing each to communicate only with a limited number of neighboring agents. This setup encourages agents to work toward an optimal solution that aligns with their neighbors' outcomes.

[C3] Learning communication strategy: The aforementioned challenge presupposes that chatbots strive for a consensus that yields the optimal policy. However, the specific information necessary for an agent to learn this policy is not always evident. Efficient negotiation through the exchange of critical messages allows agents to discern the content, timing, and recipients of their communications. This process also fosters the development of negotiation skills within the virtual community, independent of user-chatbot interactions.

### Solutions of positive incentive

[S1] Value function factorization: Recent advances in multi-agent value decomposition have significantly enhanced cooperative learning in complex environments. *Sunehag et al. (2018)* pioneered value decomposition networks to quantify individual agent contributions to collective rewards by factorizing the team value function into agent-specific components based on local state-action histories. Building on this, *Rashid et al. (2020)* introduced a monotonicity constraint in the mixing network to preserve the relationship between centralized and decentralized value functions, while incorporating global state information and non-linear transformations to improve solution quality. To address representation limitations and approximation errors, *Shen et al. (2022)* proposed residual functions for optimal joint policy learning. *Mei, Zhou & Lan (2023)* advanced this further by formulating optimal projection of mixing functions as a regret minimization problem over state-action value weights. In metaverse environments characterized by partial observability and policy diversity, action sequencing affects representable function classes and introduces training estimation errors. *Zhou, Lan & Aggarwal (2022)* developed a PAC-assisted framework using counterfactual predictions and a novel loss function for value decomposition. For risk-sensitive scenarios, *Shen et al. (2023)* established the risk-sensitive individual-global-max principle, aligning decentralized risk assessments with centralized policy. *Yuan et al. (2022a)* introduced task decomposition to streamline learning in complex environments through concentrated coordination.

[S2] Application of consensus concepts: In multi-agent systems, two primary scenarios apply when agents adhere to distinct policies. In the first, agents possess knowledge solely of their states and rewards. Conversely, in the second, a global state is visible to all, with each agent receiving a unique local reward. *Cassano, Yuan & Sayed (2020)* introduced the fast diffusion for policy evaluation algorithm, merging off-policy learning with eligibility

traces and linear function approximation, enabling the evaluation of both individual and team policies within a finite dataset. This framework permits agents to exchange information with immediate neighbors; however, the lack of message aggregation may result in suboptimal policy learning. To mitigate this, *Guan et al. (2025)* proposed multi-agent communication *via* self-supervised information aggregation, developing a permutation-invariant message encoder to amalgamate common information into concise representations, thereby refining local policy. *Wang et al. (2024b)* presented the multi-agent goal imagination framework, which employs self-supervised learning to optimize representation by reconstructing and forecasting future states, thus aligning agents toward a shared objective.

In the design of consensus-reaching processes, the number of discussion rounds and the degree of agent harmony are pivotal for measuring efficiency (*Hassani et al., 2022*). To minimize communication overhead, *Zhang & Zavlanos (2024)* introduced a distributed zero-order policy optimization method, enabling agents to compute the local policy gradients necessary for policy function updates. *Lu et al. (2021)* conceptualized the consensus-reaching process as a distributed constrained Markov decision process involving networked agents. They proposed a decentralized policy gradient approach that optimizes policies based on this model over a network.

[S3] Efficient self-structured negotiation: In the study of communication strategies, *Mordatch & Abbeel (2018)* explored a joint reward scenario where agents are aware of each other's positions and messages, sharing their policies and action spaces and utilizing symbols from a predefined vocabulary to denote their actions. However, the indiscriminate sharing of parameters can impede exploration and diminish performance. To counteract this, *Yang et al. (2022)* introduced an innovative framework that learns dynamic subtask allocation using clustering agents with analogous skill sets. *Li et al. (2023a)* proposed a hybrid framework, integrating an evolutionary algorithm with MARL to enhance coordination. Evolutionary Algorithm drives population evolution through crossover and mutation, offering a variety of experiences to MARL, which in turn refines its policies for further optimization.

In virtual communities with fluctuating numbers of chatbots, uniform message sharing or targeted communication learning may face scalability challenges in MARL communication. To overcome these issues, *Jiang & Lu (2018)* developed the attention-based targeted observation communication algorithm, which incorporates an attention mechanism to decide when to integrate information from other agents. Agents establish communication groups, maintaining a majority of consistent collaborators when they opt to communicate. Additionally, *Yuan et al. (2022b)* introduced the multi-agent incentive communication framework, enabling agents to craft incentive-laden messages that can sway the value functions of other agents, fostering effective collaboration. *Guo, Shi & Fan, 2023* proposed the transformer-based email mechanism, a framework where agents engage in local communication, sending messages exclusively to observed agents, rather than modeling the entire agent population.

Chatbot failures risk collapsing entire virtual communities, making comprehensive pre-release testing across all relevant device types critically important. As chatbot scale

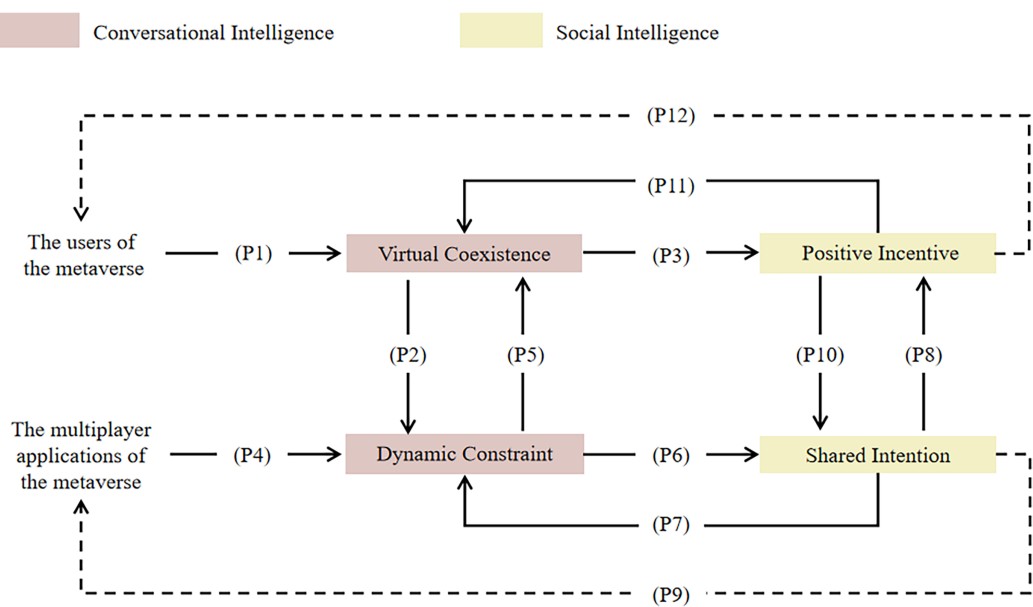

**Figure 4 Interrelationships among the elements.** The interaction among four pivotal elements within chatbots' conversational and social intelligence in the metaverse is crucial. These elements comprise virtual coexistence, dynamic constraint, shared intention, and positive incentive. The influence of virtual coexistence and dynamic constraint is primarily driven by immediate user interactions and the interplay of multiplayer applications. Conversely, shared intention and positive incentive contribute to significant long-term influences on user engagement and the collaborative efficacy of various applications. The interplay of these elements, both in the short-term, significantly shapes the user experience within the metaverse.

increases, training data demands escalate, compelling practitioners to automate and outsource data curation for state-of-the-art performance. This absence of human supervision in data collection introduces security vulnerabilities: manipulated training data can control and degrade downstream model behaviors (*Goldblum et al., 2023*). For instance, adversarial samples significantly contribute to failures. *Zhu et al. (2024)* demonstrated this vulnerability by deploying multi-level adversarial textual attacks against LLM prompts, revealing contemporary models' fragility. *Zhou et al. (2022)* reframed adversarial analysis through the lens of advanced persistent threats, modeling attacks as interconnected systems where strategically similar attack clusters form components. This systemic view enables identification of cross-stage defensive relationships, facilitating integrated frameworks that combine multiple defenses to enhance chatbot robustness.

# DISCUSSIONS

## Interrelationships among chatbots' social capabilities

In the preceding sections, we classified the social capabilities of chatbots into four key elements and examined the interdependent objectives that either promote or supplement one another. This section introduces a theoretical framework that outlines these interconnections. The aim is to elucidate the inherent relationships within potential

solutions, highlighting the complexity of chatbot development for virtual communities within the metaverse. As depicted in Fig. 4, the boxes represent the key elements, with different colors signifying their respective capabilities. Solid lines represent ten short-term influences, whereas dotted lines denote two long-term influences. The notion of endowing chatbots with social capabilities is not one of replicating human traits. However, as chatbots advance in conversational and social intelligence, they become more adept at forming IRCs with users.

- Virtual coexistence: Our chatbot model demonstrates that virtual coexistence enhances the interaction between chatbots and users (P1), acknowledging the complexity of managing conversational status in multi-participant and open-domain dialogues. Proactive conversation, grounded in contextual understanding, enhances chatbot proactivity in establishing dynamic constraint (P2), thereby demonstrating attentiveness to participants. This element also bolsters positive incentive (P3), as more communicative chatbots encourage greater user engagement in user–chatbot interactions.

- Dynamic constraint: Dynamic constraint is rooted in the multiplayer environment of the metaverse (P4), addressing challenges arising from multimodal interactions, information overload, and cross-domain tasks. Advanced multimodal information processing, robust knowledge management, and cross-domain adaptability strengthen the efficacy of dynamic constraint. This element parallels virtual coexistence (P5) by ensuring dialogue consistency through the derivation and recall of information from previous interactions. Furthermore, dynamic constraint reflects shared intention (P6), as chatbots can recall common actions or events across sessions.

- Shared intention: Shared intention impacts dynamic constraint (P7) by facilitating the exclusion of external participants, thereby enhancing group cohesion. For instance, chatbots can identify potential interlopers through improved language comprehension and generation capabilities, leveraging large language models (LLMs). Shared intention also supports positive incentive (P8), as it enables chatbots to autonomously plan and communicate, effectively applying policies from a reserved pool of positive initiative. This element can refine the multiplayer experience in the metaverse (P9), with chatbots serving as tireless work assistants, contingent on addressing issues like prompt brittleness and hallucination in future research.

- Positive incentive: Positive incentive influences shared intention (P10), as it can be used as a strategy to punish requests diverging from shared intention by reducing rewards while preserving interactive continuity. Positive incentive also influences virtual coexistence (P11), as it can help chatbots avoid multi-role conflicts and reduce the negative effects of dynamic environments. Positive incentive can make users and chatbots more interactive (P12) by providing the capability to respond effectively to group collaboration.

## Future research directions

Research on chatbots, particularly in the metaverse, addresses a critical priority, as their integration remains an emerging area of study. The challenges identified for chatbots align with those observed in human-human interaction. However, the metaverse introduces unique interactive environments and technologies that shape the development of chatbots. While bodily co-presence (*e.g.*, face-to-face encounters) fosters intense and energizing interactions, some scholars argue that low-intensity, technology-mediated interactions can also be socially effective due to their accessibility and frequency (*Johannessen & Collins, 2024*). Thus, we maintain an optimistic outlook on chatbots as members of virtual communities in the metaverse. This study outlines prospective research directions, emphasizing key areas for chatbot integration in the metaverse.

- Enhancing multimodal socio-emotional intelligence. Significant improvements are needed in chatbots' ability to perceive, interpret, and express socio-emotional cues through multimodal integration (text, speech, facial expressions, and environmental context). Advancements in emotion recognition and generation systems—such as graph-based dialogue state representations and diffusion models—are essential for synchronized, context-aware emotional expressions. Strengthening Theory-of-Mind capabilities through mental state graphs can help track user personas, beliefs, and intentions more effectively. Additionally, aligning responses with diverse social norms and cultural nuances ensures inclusive interactions in global virtual communities.

- Establishing effective human-chatbot teaming capabilities. Metaverse chatbots must function as effective "co-participants" in interaction rituals, requiring mutual predictability, directability, and common ground maintenance. Transparent decision-making processes (*e.g.*, policy gradient methods) can enhance user trust, while dynamic autonomy adjustments allow real-time customization (*e.g.*, modifying response styles in virtual education). Maintaining common ground ensures alignment with evolving user contexts, and adaptive goal negotiation frameworks enable flexible collaboration in scenarios like virtual project management.

- Building robust technical foundations and ensuring ecosystem integration. Overcoming technical limitations is crucial for scalable, safe deployment in the metaverse. Key challenges include improving interpretability in deep neural dialogue models and mitigating hallucinations *via* hybrid neural-symbolic approaches. Addressing data scarcity through LLM-driven augmentation is vital for niche domains. Explicit safety constraints (*e.g.*, signaling limitations) prevent coordination failures in high-stakes interactions. Evaluation should extend beyond accuracy to user satisfaction and task success. Standardized, schema-driven design principles are needed for cross-platform compatibility in applications ranging from customer service to collaborative problem-solving.

- Advancing scalable multi-agent collaboration frameworks. Future research should prioritize MARL frameworks to manage the metaverse's dynamic, large-scale interactions. Techniques such as mean-field MARL can effectively model aggregated

user behavior in densely populated environments (*e.g.*, social hubs), while Q-function factorization enables the decomposition of complex joint tasks, facilitating seamless collaboration among chatbots, virtual assistants, and NPCs in activities like virtual events or co-creation projects. Additionally, integrating MARL with game theory, social choice theory, and distributed optimization could enhance interaction models, improve strategic behavior handling, and optimize coordination mechanisms for complex multi-agent tasks.

## Limitations

This study has several limitations. First, as the survey focused exclusively on disembodied, text-based chatbots, literature concerning embodied robots and speech-based conversational agents was excluded. We acknowledge that incorporating these modalities can enhance chatbots' social capabilities, particularly for traits significantly influenced by physical representation, tone, and accent (*e.g.*, identity, politeness, and reasoning). However, embodiment and multimodal interaction introduce additional complexities (*e.g.*, speech synthesis or facial expression simulation) that lie beyond this study's scope and require more advanced technological capabilities. Furthermore, this exclusion may have resulted in the under-representation of specific research domains. For instance, a substantial body of work within human-robot interaction, particularly in the humanoid robotics discipline (*e.g.*, *Tong & Haotian Liu (2024)*, *Rawal & Stock-Homburg (2022)*), investigates facial emotion expression and is therefore not represented here. Similarly, research within design engineering (*e.g.*, *Spatola & Wudarczyk (2021)*, *Lawrence et al. (2025)*) has extensively examined specific social attributes and behaviors of robots, focusing primarily on social rather than functional aspects of interaction, and is consequently underrepresented in this study.

Second, because the concept of social capability lacks consensus within the literature and chatbots have been studied across diverse domains, their application within specific subfields may have been overlooked. To mitigate this limitation, we incorporated multiple synonyms into our search strategy and leveraged Google Scholar and arXiv for its broad interdisciplinary coverage.

Finally, the conceptual model of social capabilities was derived through a qualitative coding process informed by theories such as IRC. Like any qualitative method, this approach relies on researcher interpretation of interactive phenomena. To address this limitation, the researchers discussed social capabilities and categories in expert forums until consensus was reached. Both the framework and the relationships among social capabilities were based on findings from the surveyed studies.

## CONCLUSIONS

Although chatbots demonstrate significant efficacy in handling metaverse-related tasks, their social capabilities—essential for fostering virtual communities—remain understudied. We propose an analytical framework, grounded in the IRC theory, to standardize the evaluation of chatbots' social capabilities. By aligning these capabilities

with the core components of the IRC theory, our framework facilitates a systematic assessment of their conversational and social intelligence. We further identify and elaborate on four key elements—virtual coexistence, dynamic constraint, shared intention, and positive incentive—to highlight challenges and propose potential solutions.

Proactive chatbots play a pivotal role in virtual coexistence by recognizing emotions, engaging in supportive dialogue, and inferring user intentions to provide context-aware services. They must handle both single-turn and multi-turn interactions seamlessly, foster trust through anticipatory responses, and ensure communicability—especially in initial interactions—to guide users effectively. Future research could draw on metaverse literature to develop intelligent interactive systems capable of minimizing user misunderstandings. The metaverse generates vast amounts of multimodal data characterized by high variety, velocity, and veracity, presenting challenges for dynamic constraint management. MML could enhance emotion recognition and foster a sense of belonging, while KGs may serve as dynamic knowledge bases for social network representation and complex reasoning. TL could enable chatbots to generalize across constrained domains without requiring in-domain data. Despite their autonomy, LLM-based chatbots exhibit performance gaps compared to avatars, particularly in long-term reasoning, decision-making, and instruction adherence—issues that warrant further investigation. Successful interaction rituals require chatbots to provide positive incentives through multi-agent mechanisms: Value function factorization quantifies individual contributions to global rewards. Consensus idea application limits communication to sparsely connected networks. Efficient self-structured negotiation enables collaborative learning and strategy adaptation.

This study also presented the interdependencies among these elements and proposed future research directions for developing chatbot-enhanced metaverse environments. Our analysis revealed that: (1) virtual coexistence improves user-chatbot interactions, enforces dynamic constraints through proactive dialogue, and strengthens positive incentives for engagement; (2) dynamic constraints, inherent in multiplayer metaverse settings, maintain conversational consistency while reflecting shared intentions through common action recall; (3) shared intentions reinforce group cohesion through dynamic constraints, enable autonomous planning *via* positive incentives, and optimize multiplayer experiences; and (4) positive incentives regulate shared intentions by addressing divergent requests, affect virtual coexistence to avoid multi-role conflicts, and enhance interactivity in group collaboration. Key areas for future investigation include: developing advanced multimodal socio-emotional intelligence, establishing effective human-chatbot teaming capabilities, building robust technical foundations and ensuring ecosystem integration, and advancing scalable multi-agent collaboration frameworks. This study provides both theoretical and practical contributions for optimizing chatbot efficacy within the metaverse.

## ACKNOWLEDGEMENTS

The authors would like to thank all academic advisors of the Faculty of Data Science at the City University of Macau for their insightful comments.

### Funding

This research is supported by NSFC-FDCT under its Joint Scientific Research Project Fund (Grant No. 0051/2022/AFJ), China & Macau S.A.R. The funders had no role in study design, data collection and analysis, decision to publish, or preparation of the manuscript.

### Grant Disclosures

The following grant information was disclosed by the authors:
NSFC-FDCT under its Joint Scientific Research Project Fund: 0051/2022/AFJ.

### Competing Interests

The authors declare that they have no competing interests.

### Author Contributions

- Yan Yan conceived and designed the experiments, analyzed the data, performed the computation work, prepared figures and/or tables, authored or reviewed drafts of the article, and approved the final draft.
- Mengjuan Fan performed the experiments, performed the computation work, prepared figures and/or tables, and approved the final draft.

### Data Availability

This article is a literature review.

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
