# Peer review of "Chatbots integrated in the metaverse: a conceptual model based on interaction ritual chain theory"

_PeerJ Computer Science, doi:10.7717/peerj-cs.3235_

## Round 0.1 · original submission · Major Revisions

**Language Note:** When you prepare your next revision, please either (i) have a colleague who is proficient in English and familiar with the subject matter review your manuscript, or (ii) contact a professional editing service to review your manuscript. PeerJ can provide language editing services - you can contact us at [email protected] for pricing (be sure to provide your manuscript number and title). – PeerJ Staff

Reviewer 1 ·

Basic reporting

- The manuscript is written in generally professional English, though there are a few minor issues with clarity and grammar that should be revised for improved readability and precision.

- The literature review does touch on relevant studies but lacks depth in situating the current work in the broader context. Several important prior surveys and reviews in the chatbot field are not adequately acknowledged or compared. Including such comparisons would strengthen the relevance and contextual framing.

- The structure of the article is consistent with a review paper, but essential components for systematic reviews are missing. In particular, as noted in line 317, the paper identifies itself as a survey/review article. Therefore, it should comply with the PRISMA (Preferred Reporting Items for Systematic reviews and Meta-Analyses) guidelines.

Actionable suggestions:
- Include a PRISMA flow diagram and checklist, as outlined in the official PRISMA documentation https://www.prisma-statement.org/.

- Add a comparative table to clearly highlight how this review differs from and adds to existing surveys in the chatbot domain (e.g., those found in DOI: 10.1007/s10639-024-12924-4, 10.1016/j.ijer.2024.102461).

- Visual aids are minimal. The inclusion of figures such as keyword trends, frequency of publications per year, and thematic analysis would greatly enhance clarity and engagement. Consider using tools such as treemaps, heatmaps, Litmaps, or VOSviewer to enrich the visual presentation of the data.

Experimental design

- The article topic falls within the journal’s aims and scope and is appropriate for a review/survey type submission.

- The methodology, as presented, does not follow a systematic or standardized approach. There is no clear explanation of how the literature was selected, filtered, or analyzed—an essential requirement for survey papers.

- Recommendation: Clearly describe the search strategy, inclusion/exclusion criteria, and categorization methods used in the review process, as stated in https://www.prisma-statement.org.

- The article does not currently reflect a comprehensive and unbiased coverage of the literature. More attention should be given to ensuring diverse representation of sources and viewpoints.

- The organization of content is logical, but could benefit from more structured subheadings or thematic segmentation that aligns with the review goals.

- Citation practices are generally acceptable, but some claims would benefit from stronger referencing, especially when discussing conceptual models or evaluation frameworks.

Validity of the findings

- Since the paper is a review, original findings are limited. However, the synthesis of previous work lacks clear argumentation about the insights derived from the reviewed literature. The conclusion should better tie back to the research questions or motivations stated in the introduction.

- The conceptual model introduced is underdeveloped. Notably, it lacks any discussion of evaluation metrics: (1) How would the proposed model be implemented in the metaverse? (2) What are the standard evaluation metrics in this context? (3) What would be considered optimal values for these metrics? These are critical to the utility and applicability of the model and should be addressed.

- The conclusion section should include:
(1) A more detailed discussion on unresolved questions, gaps in the literature, and (2) Future research directions that could benefit the field.

- Actionable suggestion: Introduce a structured summary or table outlining key insights, unresolved challenges, and opportunities identified through the survey.

Reviewer 2 ·

Basic reporting

The article is written in clear, professional English, with a coherent overall structure. The introduction provides the necessary context and adequately justifies the choice of theoretical framework ( Interaction Ritual Chains ). The state of the art is well documented and up-to-date, with relevant references up to 2024. However, there are some aspects that could be improved, as detailed below:
• The abstract does not highlight the relevance of the study to professional practice or the academic community, which limits its potential impact. It is recommended that the abstract be expanded to explicitly include the applicability of the proposed model and its usefulness in real-world metaverse contexts (education, commerce, healthcare, etc.).
• The article presents a single research question, but refers to “ Research questions ” in the plural.
• The objectives of the study are not clearly stated, making it difficult to align the introduction, development, and conclusion.
• Some sections of the text contain excessively long and dense paragraphs, which should be better broken down and have greater structural clarity.
• Table 1 (relationship between IRC model elements and chatbot capabilities ) is interesting, but it would be better to integrate it into the discussion and highlight its practical implications.

Experimental design

Interaction Ritual Chains theory is original and justifies the article's contribution. However, I consider several methodological weaknesses:
• Although the use of a PRISMA-based strategy is mentioned, the study's methodological approach (qualitative, documentary, or conceptual) is not clearly detailed. Given that this is a desk review with theoretical development, this should be explicitly stated from the outset.
• The methodology is described ambiguously, without detailing inclusion and exclusion criteria for the reviewed studies, nor the analysis phases of the selected literature.
• The article does not formulate research objectives, either general or specific. This is problematic, as it prevents assessment of whether the study achieves its objectives and weakens the coherence between the approach, development, and conclusions.
• Neither includes hypotheses, which wouldn't be necessary in a documentary review, but there should at least be a clear statement of objectives that provides direction for the work. In its current form, it's not clear what the model is trying to demonstrate or construct beyond the model formulation.
3.

Validity of the findings

The proposed conceptual model is relevant, original, and theoretically sound. The tables presenting challenges and solutions add value and organize the literature reviewed well. However, significant weaknesses are noted:
• Although a research question is posed, there is no structured feedback on it in the discussion or conclusions.
• The discussion section overlaps with the results and does not constitute a critical reflection or systematic comparison with previous studies. A discussion that interprets the findings in light of the literature reviewed and explicitly addresses the research question is lacking.
• The conclusions are excessively brief and fail to adequately summarize the findings of the work. They also fail to offer clear projections of future lines of research or a critical assessment of the proposed model.
• Overall, the article needs a structural reworking that better connects the introduction (with its question), the theoretical model, the analytical results, and the conclusions.

Additional comments

The article addresses an emerging and relevant topic, using an innovative approach that combines data science, human-computer interaction, and sociological theory.
On the other hand, the connection between the metaverse, chatbots, and IRC theory is well-founded and can have an impact if the structural aspects mentioned are reinforced.
Finally, it is suggested to include a final section with practical or design recommendations aimed at professionals or researchers, which would increase its applicability and interest to a broad audience.

Reviewer 3 ·

Basic reporting

-

Experimental design

The article is based on a literature review. The review is quite comprehensive. As such, there is no primary data collection/ analysis.

Validity of the findings

A qualitative study with a small sample might be conducted to validate the findings.

Additional comments

In spite of the lengthy discussion, the conclusion is very short. It can be improved.

·

Basic reporting

The manuscript “Chatbots integrated in the metaverse: a conceptual model based on Interaction Ritual Chains (IRC) theory” tackles a timely intersection of social theory and conversational AI, yet it needs substantial revision before it can be recommended for publication.

The prose is broadly understandable but too often weighed down by long, multi-clause sentences—especially in the emotion-recognition discussion—so a professional copy-edit aimed at shorter sentences, consistent tense, and idiomatic phrasing would markedly improve readability. Structurally, every core section is present, yet the “Background / Related works” narrative repeatedly revisits material the Results section later covers; relocating the short “Related works and contributions” subsection (around lines 268-312) to the Introduction and trimming redundant background text would reduce this overlap.
Figures and tables convey the framework well, but Figure 1 lacks a legend for its two colour bars and uses line weights that disappear in greyscale printouts. Adding a colour-blind-safe legend and thicker or patterned lines would solve the issue. Tables 2 and 3, though valuable, sprawl across several pages; shifting the full bibliographic details to supplementary material would streamline the main text. Citations include a healthy mix of recent and foundational sources, but several remain pre-prints without persistent identifiers—every entry should carry a DOI or arXiv ID, and the reference list must fully match the journal style.

Methodologically, the authors outline a four-stage, PRISMA-inspired literature search that winnows 1,273 records down to 74 studies, but the search strings, database coverage window, exclusion criteria, and risk-of-bias appraisal are missing, rendering the process irreproducible. Inter-rater reliability statistics are likewise absent. These omissions can be rectified by (i) supplying the exact Boolean queries, databases, and dates; (ii) adding a PRISMA flow diagram with numbers at each stage; and (iii) reporting a simple quality checklist plus Cohen’s κ (or similar) for screening agreement.
The IRC-based conceptual model is coherent and logically argued, yet it remains wholly theoretical. Without even a small case study, simulation, or expert evaluation, it is impossible to judge whether the four IRC-derived design elements—mutual focus, shared mood, positive incentive, and synchronisation—truly span the metaverse-chatbot design space. A concise empirical validation, such as mapping an existing metaverse chatbot onto the framework and analysing observed gaps, would greatly strengthen the paper’s contribution. The other way, the authors could bring it as a shortcoming for the conclusion section, as a limitation.

Several mappings between IRC elements and technical capabilities are also too coarse. For instance, equating “positive incentive” directly with multi-agent reinforcement learning conflates a user-facing goal with one specific implementation path. Defining clear criteria for when a capability is satisfied and suggesting concrete success metrics would help clarify these links. Security receives only a brief, late mention, despite social deception being a genuine risk in the metaverse; integrate security either as a cross-cutting dimension of every capability or as a dedicated subsection that outlines threats, mitigations, and open research questions. Finally, acronym hygiene needs improvement: terms such as MMML, MARL, and TL should be spelled out the first time they appear, preferably already in the abstract.

Experimental design

-

Validity of the findings

Findings are valid but need some polishing and providing more information, as mentioned in the first section of the review. Please see also the summary of needed amendments in the section "additional comments"

Additional comments

Major recommended actions
• Provide the complete search protocol, a PRISMA diagram, and inter-rater reliability statistics.
• Undertake a thorough language edit to shorten sentences and correct minor grammatical issues.
• Please provide a summary of all the results in a concise manner, e.g., a table or a figure. There is Figure 3, but it seems to be too general regarding the elements written in the Results section.

Minor actions
• Improve figure legends and ensure graphics remain clear in greyscale.
• Move oversized tables (e.g., Tables 2–3) to supplementary material.
• Integrate the security discussion into the main framework.
• Define all acronyms at first use.
The topic is important and the conceptual synthesis promising, but greater methodological transparency, at least one empirical touchstone, and improved presentation are required for the manuscript to reach its potential.

---

## Round 0.2 · accepted · Accept

Both reviewers have recommended acceptance of your manuscript. Reviewer 1 has no further comments, and Reviewer 2 has confirmed that the revisions satisfactorily addressed all prior observations, improving clarity, methodological soundness, and coherence.

Accordingly, I am pleased to inform you that your paper is accepted for publication.

Congratulations, and thank you for your valuable contribution.

Reviewer 1 ·

Basic reporting

-

Experimental design

-

Validity of the findings

-

Additional comments

OK. No more comments.

Reviewer 2 ·

Basic reporting

No comments. The revised manuscript meets the journal’s standards and has satisfactorily incorporated the improvements requested in the first review round.

Experimental design

-

Validity of the findings

-